# Normal Difference Vegetation Index Simulation and Driving Analysis of the Tibetan Plateau Based on Deep Learning Algorithms

**Xi Liu [1], Guoming Du [1,\*] , Haoting Bi [2], Zimou Li [3] and Xiaodie Zhang [4]**

[1] School of Geography and Planning, Sun Yat-sen University, Guangzhou 510006, China; liux663@mail2.sysu.edu.cn
[2] School of Chemical Engineering and Technology, Sun Yat-sen University, Zhuhai 519082, China; biht@mail2.sysu.edu.cn
[3] School of Science, Guangdong University of Petrochemical Technology, Maoming 525000, China; li76939088@gmail.com
[4] Beidou Research Institute, South China Normal University, Foshan 528225, China; 2023025241@m.scnu.edu.cn
[\*] Correspondence: eesdgm@mail.sysu.edu.cn; Tel.: +86-18024530675

**Abstract:** Global climate warming has profoundly affected terrestrial ecosystems. The Tibetan Plateau (TP) is an ecologically vulnerable region that emerged as an ideal place for investigating the mechanisms of vegetation response to climate change. In this study, we constructed an annual synthetic NDVI dataset with 500 m resolution based on MOD13A1 products from 2000 to 2021, which were extracted by the Google Earth Engine (GEE) and processed by the Kalman filter. Furthermore, considering topographic and climatic factors, a thorough analysis was conducted to ascertain the causes and effects of the NDVI's spatiotemporal variations on the TP. The main findings are: (1) The vegetation coverage on the TP has been growing slowly over the past 22 years at a rate of 0.0134/10a, with a notable heterogeneity due to its topography and climate conditions. (2) During the study period, the TP generally showed a "warming and humidification" trend. The influence of human activities on vegetation growth has exhibited a favorable trajectory, with a notable acceleration observed since 2011. (3) The primary factor influencing NDVI in the southeastern and western regions of the TP was the increasing temperature. Conversely, vegetation in the northeastern and central regions was mostly regulated by precipitation. (4) Combined with the principal component analysis, a PCA-CNN-LSTM (PCL) model demonstrated significant superiority in modeling NDVI sequences on the Tibetan Plateau. Understanding the results of this paper is important for the sustainable development and the formulation of ecological policies on the Tibetan Plateau.

**Keywords:** Tibetan Plateau; NDVI; deep learning; climate change; human activities; GEE

## 1. Introduction

According to the global climate change report released by the IPCC, the average surface temperature of the Earth has risen by 0.74 °C over the past few decades, and there has been an increase in extreme precipitation events globally [1]. As a result of the dual impacts of climate change and human activities, the natural environment has also shown different degrees of alteration [2]. In nature, vegetation is an indispensable part of terrestrial ecosystems. It plays a vital role in maintaining ecosystem stability, and, as an "indicator" of regional environmental change, vegetation has a feedback mechanism for climate change [3]. Climate change affects the growth and distribution of vegetation by directly influencing its material exchange [4]. The Tibetan Plateau, as the largest and highest plateau in the world, is called the "Roof of the World" [5]. As a result of climate warming, the TP, which is influenced by different climatic zones, has undergone significant changes, and one of the most significant responses is the abnormal increase in temperature [6]. The response of the

TP to climate change is not homogeneous, and temperature changes are more pronounced at higher altitudes [7]. Researchers also show that mountains far from the influence of urbanization and topographic shading provide a more unbiased record of climate than cities [8]. As one of the most sensitive regions to climate change, the TP has received much attention, which makes it a suitable entry point for global climate change research. Therefore, analyzing the growth and distribution of vegetation on the TP and its response to climate change is significant to studying global warming and ecological regulation.

Monitoring the vegetation indices by remote sensing is an effective way to study the distribution pattern of vegetation [9]. Normal Difference Vegetation Index (NDVI) is a reliable indicator for assessing ecological status and is especially suitable for large-scale regions with limited vegetation coverage [10]. Consequently, employing NDVI as an indicator for monitoring vegetation dynamics on the TP offers notable benefits [11]. In recent decades, numerous studies have shown that the vegetation on the TP has been characterized by "Overall improvement, Local degradation, with significant spatial heterogeneity" [12,13]. Piao [14] analyzed the trend of NDVI in China from 1982 to 1999 and found that more than 80% of the area showed an increasing trend, with significant spatial and seasonal heterogeneity. Other studies conducted revealed that the NDVI exhibited a modest upward trajectory over the TP [13,15].

Furthermore, it was observed that there were discernible patterns of variation across various latitudes/longitudes, seasons, and climatic zones [16,17]. The aforementioned studies indicate that the complex topography and diverse climate types have likely led to the heterogeneity of the distribution and variation of NDVI [18]. Researchers have conducted extensive and in-depth studies on the reflection of vegetation change characteristics in recent years. One-way linear regression and the Theil–Sen trend method are frequently used to determine the slope of the index, which reflects the change (improvement or degradation) of the vegetation based on the positivity or negativity of the slope [19]. Meanwhile, statistical tests such as the $p$-test, $t$-test, or Mann–Kendall test were employed to evaluate the significance of the trend results for a more in-depth study of vegetation dynamics [20].

Nevertheless, the current research mostly concentrates on the historical changes of vegetation indices and disregards their forthcoming evolutionary patterns. Jiang [21] examined the consistency of changes in vegetation cover in the Loess Plateau with the Hurst index. Omer [22] employed a combination of two machine-learning techniques, Support Vector Regression (SVR) and Artificial Neural Network (ANN), for the inversion of LAI. This approach yielded substantial validation accuracy, as indicated by an R-squared value of 0.75. In studies where different models were used to predict the global GPP, researchers found that the LSTM model consistently had the lowest RMSE [23]. Chen [24] used an LSTM model with temperature and precipitation data as input variables to predict future changes in vegetation NDVI. Jin [25] discovered that, in northwest China, the impact of humidity on NDVI should not be disregarded and should be included in the deep learning algorithm's inputs. Deep learning models provide superior accuracy and efficiency in addressing ecological multidimensional data compared to traditional machine learning techniques [26].

Currently, more and more researchers are focusing on the response mechanism of vegetation to climate change. Researchers frequently examine the association between NDVI and climate factors (temperature, precipitation, humidity, etc.) through Pearson's correlation coefficient, partial correlation coefficient, multiple linear regression, etc. to quantify the response of vegetation to climate change [27]. Various factors influence vegetation growth, broadly categorized as natural and anthropogenic forces [28]. Human activities often exert control over the impact of climate on vegetation growth via means such as deforestation, reforestation of agricultural land, or vegetation restoration [27]. Residual trends are often used to quantify the impacts of human activities [19]. In addition, researchers demonstrated that land cover type, altitude, and hydrothermal conditions in the study area affect the response mechanism of vegetation [29].

Additionally, the significance of these correlations was examined to ascertain the approximate distribution of the dominant factors [30]. This method has a significant limitation: it can neither determine the distribution of the dominant factors nor quantify the proportion of each dominant factor in the study area. Zhao used a geodetector approach to measure the explanatory degree of each factor affecting the NDVI change in Shaanxi Province to quantitatively assess each factor's contribution to the change in vegetation [31]. Unfortunately, this method could not clarify the spatial distribution of the dominant factors [32]. Although previous studies have, to a certain extent, derived the distributional characteristics of NDVI on the TP and its response relationship with climate factors, they have not considered the influence of topography on the response mechanism. In addition, most of the studies only chose temperature and precipitation as the driving factors, and did not elucidate the distribution of climate-dominant factors. By comparing the effectiveness of multiple linear regression, support vector machine, generalized additive model, and random forest model in explaining changes in vegetation cover, Qiao found that the random forest model has a lower fitting error and is more powerful in explaining changes in vegetation cover [33]. In light of the aforementioned information, we suggest using a random forest algorithm to ascertain the influence of individual drivers. This would be achieved by evaluating the Gini coefficient of each decision tree, which serves as an indicator of the spatial dispersion of the primary component [34]. The random forest can accurately identify the key drivers of vegetation change by its high simulation accuracy, fast computing speed, reliable result output, and robust generalization ability [35]. Thanks to its excellent performance, random forest has shown promising potential in vegetation change attribution.

To effectively address and further explore the research gaps above, the primary purpose of our study was as follows: (1) Use GEE as a supporting platform to further explore the variation characteristics of NDVI at varying terrain zones. (2) Quantify the driving effects of climate change and human activities on vegetation growth. (3) Develop a deep learning model to simulate NDVI on the TP accurately. The results can provide a theoretical foundation and reference basis for ecological conservation and policy-making on the TP.

## 2. Study Area and Materials

### 2.1. Overview of Study Area

The Tibetan Plateau, known for its highest and widest plateau in the world, ranges from $73°19'\sim104°47'$ E and $26°00'\sim39°47''$ N. It is distributed in six provinces, covering a total area of about $2.57 \times 10^6$ km$^2$, about 26.7% of the total area of China [36]. Five representative regions (Figure 1, A~E) have been chosen based on a detailed analysis of topography, climate, and vegetation distribution features. The topography of the TP is undulating, with an average elevation of 4400 m [37]. The regions on the TP with altitudes ranging from 4500 to 5000 m occupy the largest part of the total territory, amounting to 32.08% (Figure 1). High-altitude regions are extensively dispersed throughout the central and western sections of the plateau. In contrast, low-altitude regions constitute a lesser fraction, primarily concentrated in the southern foothills of the Himalayas and the Qaidam basin (region E). Moreover, Moderate incline (MO) occupies the widest area, amounting to 33.24%. Steep (ST) slopes cover the least area, only 4.72%. The plateau's edge tends to be steeply sloped, especially the Hengduan Mountain (region B). The Level (LE) area is mainly located in the Qaidam Basin (E). It is worth mentioning that the Northern Tibetan Plateau (region D), despite its higher elevation, has a gentler slope.

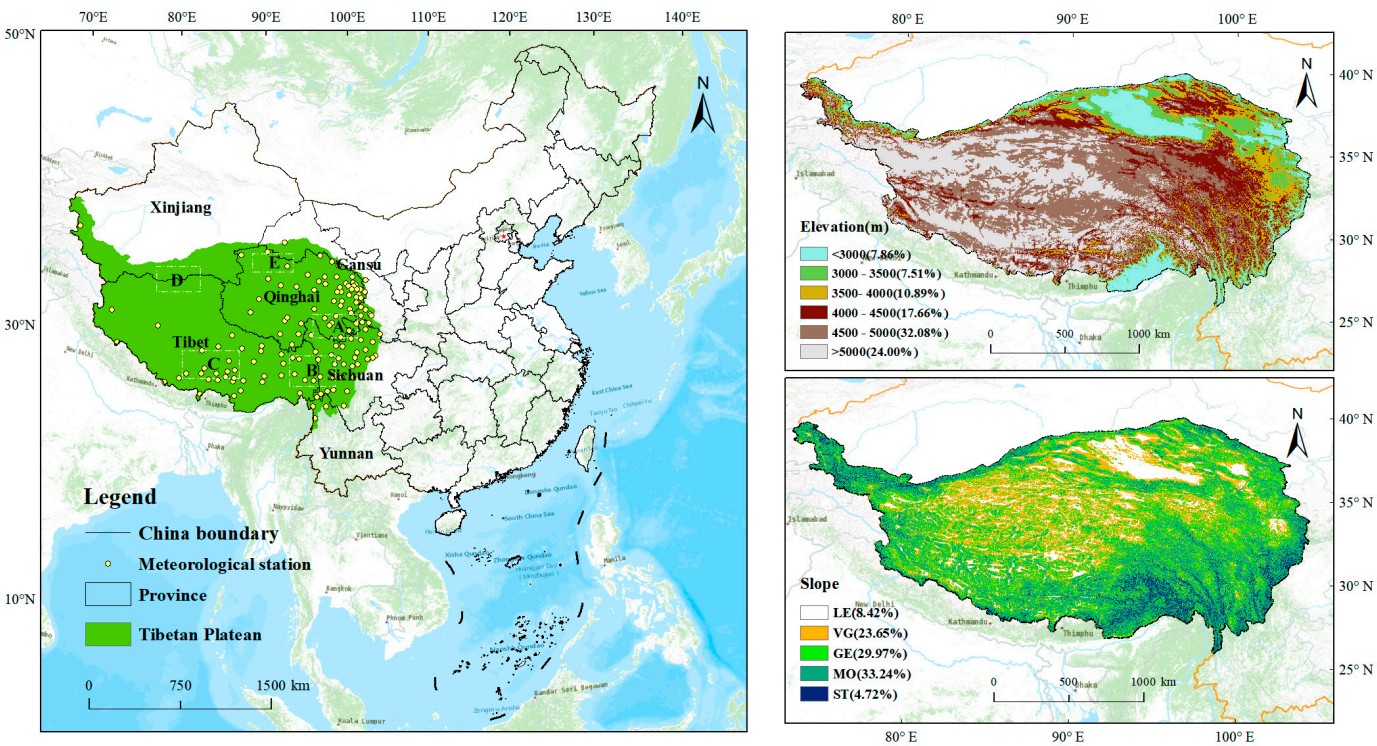

**Figure 1.** Overview of the study area. Representative regions A~E are Sanjiangyuan, Hengduan Mountains, Southwestern Tibetan Plateau, Northern Tibetan Plateau, and Qaidam Basin, respectively. Elevation grading criteria are low elevation (<3500 m), middle elevation (3500~5000 m), high elevation (>5000 m); slope grading criteria are Level (0~0.5°, LE), Very gentle incline (0.5°~2°, VG), Gentle incline (2°~5°, GE), Moderate incline (5°~15°, MO), Steep (>15°, ST).

*2.2. Data Sources and Processing*

The NDVI original dataset used in this paper is sourced from a public data archive provided by Google Earth Engine (GEE). MOD13A1 is a data product released by NASA, which has been processed with basic geometric and atmospheric corrections, etc., with a spatial resolution of 500 m and a temporal resolution of 16 days [38]. GEE is an open-source cloud computing platform for global-scale storage management, online processing, and visualization of massive geoscience datasets [10,39]. We use GEE as a support platform to extract 483 MODIS images of the study area. Unfortunately, due to the presence of clouds etc., the NDVI often shows abnormally low values, which may seriously affect the accuracy of the result [40]. This study employed the maximum value composite (MVC) approach to generate the yearly NDVI series. The Quality Control (QC) band extraction is performed using MRT v4.x software, enabling the rapid identification of aberrant NDVI values. (Data with QC = 0 or 1 can be used directly; QC = 2 or 3 indicate that snow, ice, or cloud can potentially affect them.) Those affected data are considered noise and therefore deemed necessary to be discarded. Subsequently, the Kalman filter is employed to rebuild the pixels [41]. The NDVI dataset was successfully reconstructed, demonstrating a high level of validation accuracy by the underlying concepts.

The temperature and precipitation sources are CRU v4.05, a global dataset produced and maintained by the University of East Anglia, UK. The spatial resolution of this dataset was 0.5° × 0.5°, which is slightly insufficient for studying the climate drivers on the TP. Therefore, this study used the Delta method to downscale the raw data to a spatial resolution of 1 km. The Delta method is a robust technique for modifying regional climate data and is known for its high accuracy and excellent applicability [42]. The meteorological data from the National Earth System Science Data Center (NESSD) were subjected to cross-validation with the downscaled outcomes. In addition, the datum of the remote

sensing maps made in this paper is D_WGS_1984, and the projected coordinate system is Krasovsky_1940_Albers. The rest of the data and sources are shown in Table 1.

**Table 1.** Overview of data sources.

| Category | Index | Description (Unit) | Period | Sources |
|---|---|---|---|---|
| Terrain factors | DEM | Elevation (m) | 2021 | https://lpdaac.usgs.gov/ (accessed on 10 July 2023) |
| | Slope | Slope (°) | 2021 | Extract from DEM |
| | Aspect | Aspect (°) | 2021 | Extract from DEM |
| Vegetation Index | NDVI | MOD13A1 product | 2000~2021 | https://earthengine.google.com/ (accessed on 16 December 2022) |
| Meteorological factors | T | Temperature (°C) | 2000~2021 | https://www.uea.ac.uk/ (accessed on 5 June 2023) |
| | P | Precipitation (mm) | 2000~2021 | https://www.uea.ac.uk/ (accessed on 5 June 2023) |
| | RH | Relative humidity (%rh) | 2000~2021 | http://loess.geodata.cn/ (accessed on 5 June 2023) |
| | SR | Solar radiation (W/m²) | 2000~2021 | http://loess.geodata.cn/ (accessed on 5 June 2023) |

## 3. Research Methods

### 3.1. Distribution Index

In order to reveal the influence of topography on vegetation changes, this paper introduces the distribution index to eliminate the uncertainty of vegetation restoration evaluation caused by the difference in absolute area of topographic conditions [43]. The formula is as follows:

$$K = \frac{(S_{ie}/S_i)}{(S_e/S)} \tag{1}$$

where $S_i$ is the area of vegetation change type i, $S_e$ denotes the whole area of terrain e, $S_{ie}$ is the area occupied by i under e, and S represents the overall study area. If K > 1, the ratio of the area of a specific vegetation change type under a given terrain condition ($S_{ie}/S_i$) is found to be bigger than the ratio of this vegetation change type in the overall study area ($S_e/S$). This observation indicates that terrain e is the dominant terrain for this vegetation change type i. Moreover, it is noteworthy that an increase in the value of K corresponds to a higher degree of dominance, and vice versa.

### 3.2. Trend and Residual Analysis

The Theil–Sen median method, as a robust nonparametric statistical trend algorithm, can eliminate the interference of outliers in NDVI series [19]. We used this method to assess the trends of NDVI and climate factors. It is calculated as follows:

$$\beta = median\left(\frac{X_j - X_i}{j - i}\right), i < j \leq n \tag{2}$$

where n is the period, and here n is 22; $X_i$ denotes the NDVI in year i. The positivity, negativity, and magnitude of β can well reflect the trend of vegetation growth (improvement or degradation). Our findings indicate that NDVI exhibits stable behavior across the range of β values between −0.0005 and 0.0005. The Mann–Kendall test is a nonparametric statistical test often combined with the Theil–Sen Median to determine the trend of factors over time (Table 2) [20]. Additionally, we employ the coefficient of variation (CV) to quantify the stability of the NDVI series [44].

**Table 2.** Proportion of growing trends of vegetation on the TP.

| Class | Categories | Legend | Range | Area Percentage (%) |
|---|---|---|---|---|
| Growing Trend | Significant degradation | SID | $\beta < -0.0005; |Z| > 1.96$ | 1.14% |
| | Slight degradation | SLD | $\beta < -0.0005; |Z| < 1.96$ | 9.05% |
| | Stable | STA | $|\beta| < 0.0005$ | 21.17% |
| | Slight Improvement | SLI | $\beta > 0.0005; |Z| < 1.96$ | 33.72% |
| | Significant Improvement | SII | $\beta > 0.0005; |Z| > 1.96$ | 34.92% |
| Continuity | Continuously improvement | CI | $\beta > 0.0005; Hurst > 0.5$ | 29.83% |
| | Continuously degradation | CD | $\beta < -0.0005; Hurst > 0.5$ | 3.41% |
| | Improvement→Degradation | ID | $\beta > 0.0005; Hurst < 0.5$ | 38.82% |
| | Degradation→Improvement | DI | $\beta < -0.0005; Hurst < 0.5$ | 6.78% |
| | No significant changes | NS | $|\beta| < 0.0005$ | 21.17% |

The residual trend method, first proposed by Evans and Geerken, separated the effects of climate change and human activities on vegetation growth [45]. This is achieved by constructing a multiple linear regression model incorporating NDVI and climate factors and predicting the vegetation changes only affected by climate change. The calculation formula is as follows.

$$NDVI_{cc} = a \times T + b \times P + c \times RH + d \times SR + \varepsilon \tag{3}$$

$$\delta = NDVI_{obs} - NDVI_{cc} \tag{4}$$

In Equation (3), a, b, c, and d are the regression coefficients of NDVI with temperature, precipitation, humidity, and solar radiation, respectively. $NDVI_{obs}$ is the observed value of NDVI, and $NDVI_{cc}$ is the predicted value, and the difference between the two is regarded as the part contributed by human activities, i.e., the residual. When $\delta > 0$, human activities have a positive effect on vegetation growth. Furthermore, we evaluate the relation between NDVI and climate factors using the Pearson correlation coefficient [18].

### 3.3. Hurst Exponent

A hydraulic scientist from the UK introduced the Hurst exponent, a metric that captures the underlying long-term trend in a series [45]. Researchers have also employed this concept through extensive investigation to examine vegetation dynamics [46]. The procedure can be briefly described as follows:

1.  For a given NDVI sequence $\{\overline{NDVI_{(t)}}\}$(t = 1,2,...,n), its mean sequence can be expressed as:

$$\overline{NDVI_{(\tau)}} = \frac{1}{\tau}\sum_{t=1}^{\tau} NDVI_{(t)} \quad \tau = 1, 2, 3\ldots \tag{5}$$

2.  The accumulated deviation was:

$$U_{(t,\tau)} = \sum_{t=1}^{\tau} \left(NDVI_{(t)} - \overline{NDVI_{(\tau)}}\right) \quad 1 \le t \le \tau \tag{6}$$

3.  A range of R was specified as:

$$R_{\tau} = maxU(t,\tau) - minU(t,\tau) \quad (1 \le t \le \tau; \tau = 1, 2, 3\ldots, n) \tag{7}$$

4.  The standard deviation is:

$$S_{\tau} = \left[\frac{1}{\tau}\sum_{t=1}^{\tau} NDVI(t) - NDVI(\tau)^2\right]^{\frac{1}{2}} \tag{8}$$

5.  If $\frac{R_{\tau}}{S_{\tau}} \propto \tau^H$, it indicates the presence of Hurst phenomenon in the time series:

$$\frac{R_{\tau}}{S_{\tau}} = (c\tau)^H \tag{9}$$

6.  The Hurst exponent can be obtained by fitting the equation:

$$log(R/S)_a = a + H \times log(n) \tag{10}$$

In the formula, if 0.5 < Hurst < 1, it indicates that the future trend of NDVI is consistent with the original trend, and the closer the value of Hurst is to 1, the stronger the consistency; if 0 < Hurst < 0.5, it shows the reversal of the existing trend, i.e., inconsistency; Hurst = 0.5 indicates that NDVI will change stochastically. The combination of Hurst and Theil–Sen median trend analysis can be used to forecast the future evolution of NDVI.

*3.4. Random Forest Algorithm*

Random forest is an integrated learning algorithm developed by Breiman that combines multiple decision trees to accomplish classification and regression tasks [47]. Random forest was used to evaluate the relative importance of features [48]. The main evaluation metric was Gini index, and it could be expressed as:

$$Gini_m = \sum_{k=1}^{|y|} \sum_{k' \neq k} p_k p_{k'} = 1 - \sum_{k=1}^{|y|} p_k^2 \tag{11}$$

where y indicates y categories, and $p_k$ denotes the proportion of the kth category in node m. The variable j has an importance score at node m:

$$VIM_{jm}^{(Gini)} = Gini_m - Gini_l - Gini_r \tag{12}$$

In this study, we used temperature, precipitation, humidity, and solar radiation as features to predict the dependent variable NDVI. The feature importance scores of the random forest are stored in the "feature_importances" attribution in Scikit-learn. As the score increases, there is a corresponding increase in the contribution of the climate factors to NDVI. Scikit-learning is a Python library developed specifically for practical application in machine learning. Using the RandomForestRegressor() function from the library, a random forest model was constructed. Based on a thorough analysis of data volume and computation rate, we used 10-fold cross-validation, where the training set was divided into 10 equal parts, 9 of which were used as the training set and the remaining as the validation set.

*3.5. PCA-CNN-LSTM(PCL) Model*

The framework of the PCL model is illustrated in Figure 2. In the first segment of the model, Convolutional Neural Network (CNN) was employed to preprocess raw data [49]. The convolutional layers employed kernels to extract features, generating an initial feature matrix. Following this, a pooling layer was employed to decrease the dimensionality of the features while preserving the most significant characteristics. The CNN model enabled it to extract latent information from local variables effectively.

Additionally, we have constructed a feature selector that employs principal component analysis (PCA) to reduce the dimensionality of the multi-dimensional input variables. This process selects principal components $F_i$, which exhibit substantial information content and are mutually independent. This approach not only ensures model accuracy but also enhances computational efficiency. In this study, the selection criteria for the input principal components were as follows: The cumulative variance contribution surpasses 95% and single component's contribution exceeds 10% (Table A1).

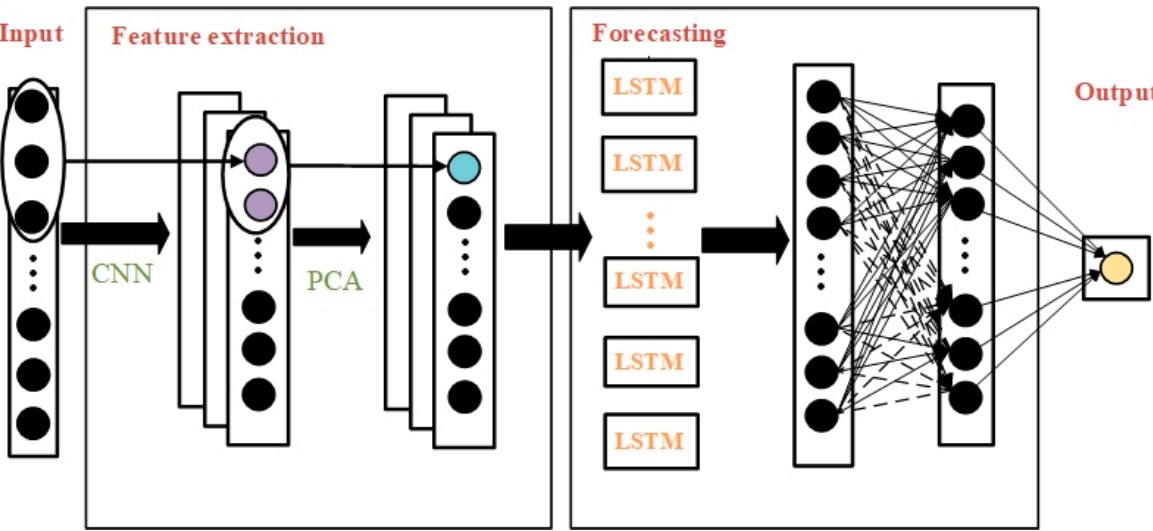

**Figure 2.** The framework of PCA-CNN-LSTM(PCL) model. The black circle is the original input data, the purple circle is the features extracted by CNN, the blue circle is the feature principal components, and the yellow circle is the output data.

The second module was LSTM layers, which extract relevant vectors for constructing time series [50]. The LSTM model is a variant of the Recurrent Neural Network (RNN) specifically designed to tackle managing long-term dependencies. The activation function Sigmoid is employed, effectively addressing the gradient explosion issue and possessing a unilateral inhibitory property [51]. Furthermore, the model incorporates the Adam optimization algorithm, which expedites weight fitting and enhances the model's robustness. This paper uses 70% of the NDVI data as the training dataset and allocates the remaining 30% as the test dataset.

## 4. Experiments and Results

### 4.1. Spatial Distribution Patterns

We employed GEE to take an average of each pixel of the NDVI dataset. The study area was predominantly comprised of regions with low and lower coverage, accounting for 41.3% and 18.9%, respectively. This region experiences a harsh climate with extremely cold and dry conditions, resulting in infertile soils and a limited capacity for vegetation growth (Figure 3a). Additionally, as longitude increases, so does the vegetation coverage. The area with NDVI > 0.6 accounts for only 28.5%, and is mainly located in (1) the northeast of TP, which has a lower elevation and gentle incline, rendering it conducive for the development of the plantation. This phenomenon fosters favorable circumstances for the proliferation of vegetation. (2) The southeast of TP is high in altitude and steep in slope but rich in biodiversity and plentiful in water resources, hence fostering thriving vegetation [52]. The area of medium coverage is characterized by a "narrow band" that divides the low coverage (left) from the high coverage (right). Moreover, the NDVI on the TP is stable, with areas of high stability primarily found in the eastern and southern regions of the plateau (Figure 3b). The distribution of moderately stable areas, however, is relatively dispersed. The regions characterized by low stability are primarily situated in the northern section of the plateau. In summary, the NDVI in TP is generally low, with noticeable spatial heterogeneity and longitudinal zonation, indicating an overall distribution pattern of "low in the northwest and high in the southeast".

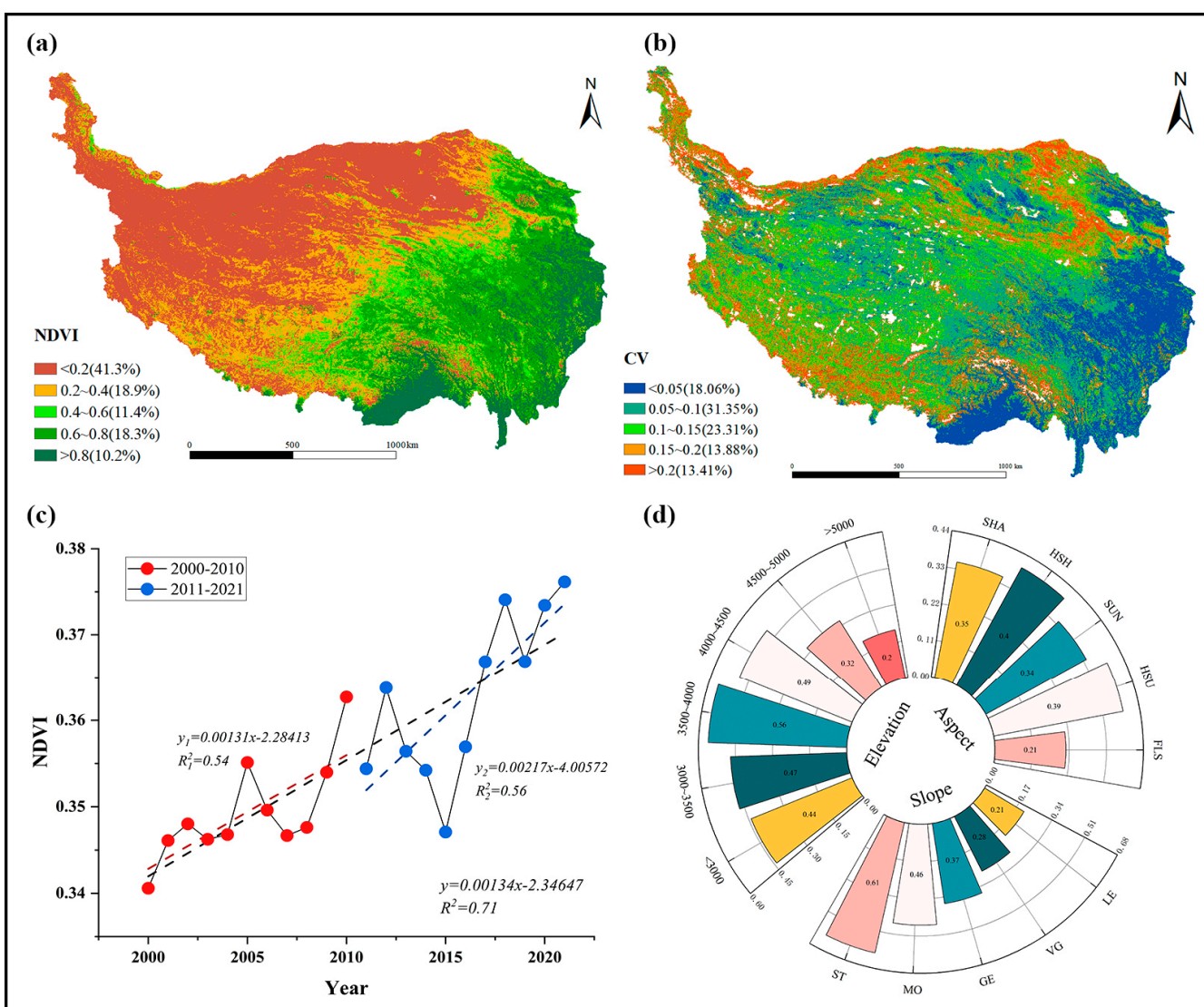

**Figure 3.** Vegetation coverage map of TP over the period 2000–2021, including (**a**) the mean NDVI, (**b**) the variation coefficient of NDVI, (**c**) interannual NDVI, and (**d**) average NDVI at varying topography zones. This paper categorized NDVI into five distinct groups: NDVI < 0.2 (lower coverage), 0.2 < NDVI < 0.4 (low coverage), 0.4 < NDVI < 0.6 (medium coverage), 0.6 < NDVI < 0.8 (high coverage), and NDVI > 0.8 (higher coverage). CV's grading criteria are CV < 0.05 (Higher stability), 0.05 < CV < 0.1 (Higher stability), 0.1 < CV < 0.15 (Moderate stability), 0.15 < CV < 0.2 (Low stability), and CV > 0.2 (Lower stability).

Based on the findings above, it has been shown that topography, a fundamental element of natural circumstances, influences the dispersion of regional hydrothermal conditions. To further research the impact of topography on vegetation coverage, we graded elevation, slope, and aspect (Figures 1 and 3d), and analyzed the distribution of the average NDVI under varying topographic conditions. Among them, the aspect was classified into five categories, which were shady aspect (0°~45° & 315°~360°, SHA), half-shady aspect (45°~135°, HSH), sunny aspect (135°~225°, SUN), half-sunny aspect (225°~315°, HSU), and flat slope (−1°, FLS). The distribution of NDVI on TP was insensitive to changes in aspect, with comparable NDVI on HSH and HSU (about 0.4), which was more significant than that on SHA and SUN (about 0.35). The maximum NDVI was found between 3500 and 4000 m, with larger values occurring at lower elevations. The NDVI dropped to 0.49 when the elevation was over 4000 m, and it continued to decline with

increasing elevation. Moreover, we found that the NDVI of TP kept rising as the slope increased, from a low of 0.21 on level ground (primarily in the Qaidam Basin) to a high of 0.61 on steep ground.

### 4.2. Temporal Variation Characteristics

Interannual variation of the average NDVI on the TP between 2000 and 2021 (Figure 3c) reveals a fluctuating growth pattern, with a growth rate of 0.0134/10a ($R^2$ = 0.71). The mean value of NDVI was 0.356, with the maximum appearing in 2021 (0.376) and the minimum appearing in 2000 (0.341). The interannual changes of NDVI had apparent phases. In the first phase: 2000~2010, the mean values of NDVI and standard deviation were 0.349 and 0.275, respectively. Additionally, the growth rate was 0.0131/10a ($R^2$ = 0.54), which was lower than the value of the entire study period. This indicated that the vegetation coverage during this phase was relatively low, exhibiting slower growth and smaller fluctuations. The subsequent period, 2011~2021, had elevated NDVI and growth rate (0.0217/10a) compared to the entire period. These findings suggest a significant acceleration in vegetation growth during this phase. It is worth mentioning that the average standard deviation had a significant increase to 0.80 from 2014 to 2017, coinciding with a pronounced fluctuation in NDVI. Overall, vegetation on the TP during the 22 years tended to improve. The observed rise in vegetation coverage over the latter decade is likely the primary reason contributing to the upward trend of NDVI in the study period [53].

### 4.3. Spatiotemporal Variation Trend

In this paper, we explored the trend of NDVI on TP from 2000 to 2021 and assessed its statistical significance (Table 2 and Figure 4). The findings showed that the trend of NDVI was in the range of −0.34 to 0.38/10a. Most of the study areas (68.65%) saw an upward trend, demonstrating significant spatial heterogeneity. Additionally, the trend steadily decreased from north to south. From the results of the Mann–Kendall test, the average Z value was 1.44, and the significance level of vegetation change was not high in general. We categorized the changing trend into five categories by using 1.96 as the critical value of Z (if Z > 1.96, it means that a significance level of 0.05 was reached). Among them, the area of significant improvement is the largest, amounting to $9.8 \times 10^5$ km$^2$, accounting for 34.92% of TP, mainly distributed in the northern part of the plateau, such as the northern Tibetan Plateau and the Qaidam basin. The vegetation degradation area only accounts for 10.19%, and is concentrated in the southwest Tibetan Plateau. In summary, the vegetation change on the TP shows a discernible pattern of "general improvement, local stabilization, and degradation located in the southwest".

The average Hurst exponent of the TP was 0.43 and the areas with Hurst < 0.5 accounted for 67.64% of the total area. This indicated that the persistence of NDVI on the plateau is relatively weak, and a trend reversal is likely to occur in the future. To further explore the future trend of the vegetation, the Theil–Sen median trend was coupled with the Hurst exponent. The results were categorized into five categories according to the rules of Table 2 about 45.6% of the TP will present an inconsistent change trend in the future, of which the improvement reversal amounts to 38.82%. It is primarily concentrated in the north-central region, which is currently at risk of vegetation degradation due to the combined effects of climate change and human activities. In addition, the consistency of the improvement trend is weaker than the degradation. The regions exhibiting consistent improvement are primarily in the Qaidam Basin and the northern TP. Conversely, the areas experiencing consistent degradation comprise the smallest proportion and are sparsely distributed in the southwestern TP.

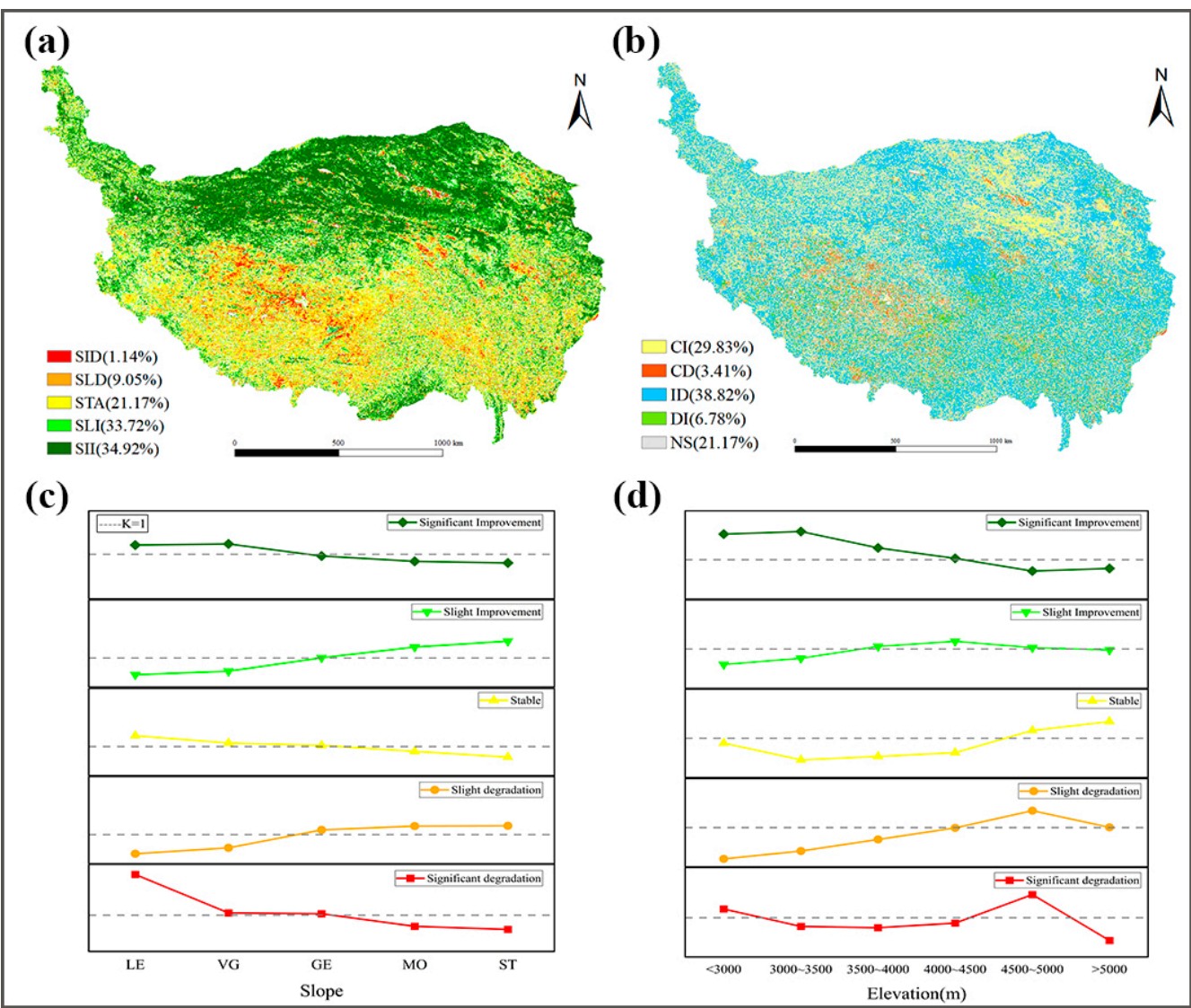

**Figure 4.** Spatiotemporal variation characteristics of NDVI from 2000 to 2021, including (**a**) the growing trend, the abbreviation principle of the legend referred by Table 2, (**b**) future trend of vegetation dynamics, terrain distribution index at varying (**c**) slope zones, and (**d**) elevation zones.

To investigate the dynamic patterns of vegetation on the TP in response to varying terrain conditions, we employed the distribution index to identify the predominant terrain conditions. In Figure 4c,d, the dotted line represents K = 1. If it is higher than the dotted line, the terrain is the dominant terrain for that particular type of vegetation change. The distribution index of vegetation improvement and degradation exhibits a symmetrical distribution pattern across each slope grade, leading to a mutual offsetting effect. Consequently, the influence of slope on the distribution index of vegetation change types is less significant (Figure 4a). Of these, LE has a more significant trend of degradation, thus highlighting the urgent need to enhance urban ecological regulations and save the indigenous flora in the area. The distribution index has prominent differentiation characteristics with altitude. Low elevation is the dominant terrain of significant improvement (1.03 < K < 1.58). The most dominant zone is found between 3000 and 3500 m with a K value of 1.58. However, as the elevation exceeds 4500 m, the value of K decreases rapidly to 0.77. Middle altitude (4000~5000 m) is the dominant terrain of slight improvement and degradation. Within this zone, the elevation at 4500~5000 m is the most dominating in terms of degradation (with a K value of 1.35 for slight degradation and 1.47 for significant degradation). Our findings

revealed that significant vegetation improvement is most pronounced at low altitudes, whereas degradation is more prevalent at medium altitudes, particularly in the range of 4500~5000 m. However, vegetation tends to stabilize at medium-to-high altitudes.

### 4.4. Correlation between NDVI and Climate Change

The Tibetan Plateau, the earth's third pole, exhibits interannual fluctuations in temperature, precipitation, humidity, and solar radiation (Figure 5a–d). The majority of the study area (69.62%) has a notable upward trajectory in temperature, with a trend of 0.104 °C/10a. Furthermore, the spatial analysis revealed a steady extension of the temperature increase trend from north to south. The cooling area is mainly distributed in the Qaidam Basin, which is at a relatively low altitude, and the atmospheric circulation may be the main reason for the temperature decline [54]. During the study period, about 55.24% of the TP experienced an increase in precipitation, with a trend of 1.80 mm/10a. The areas experiencing an increase in precipitation are primarily situated in the northeastern and southwestern TP.

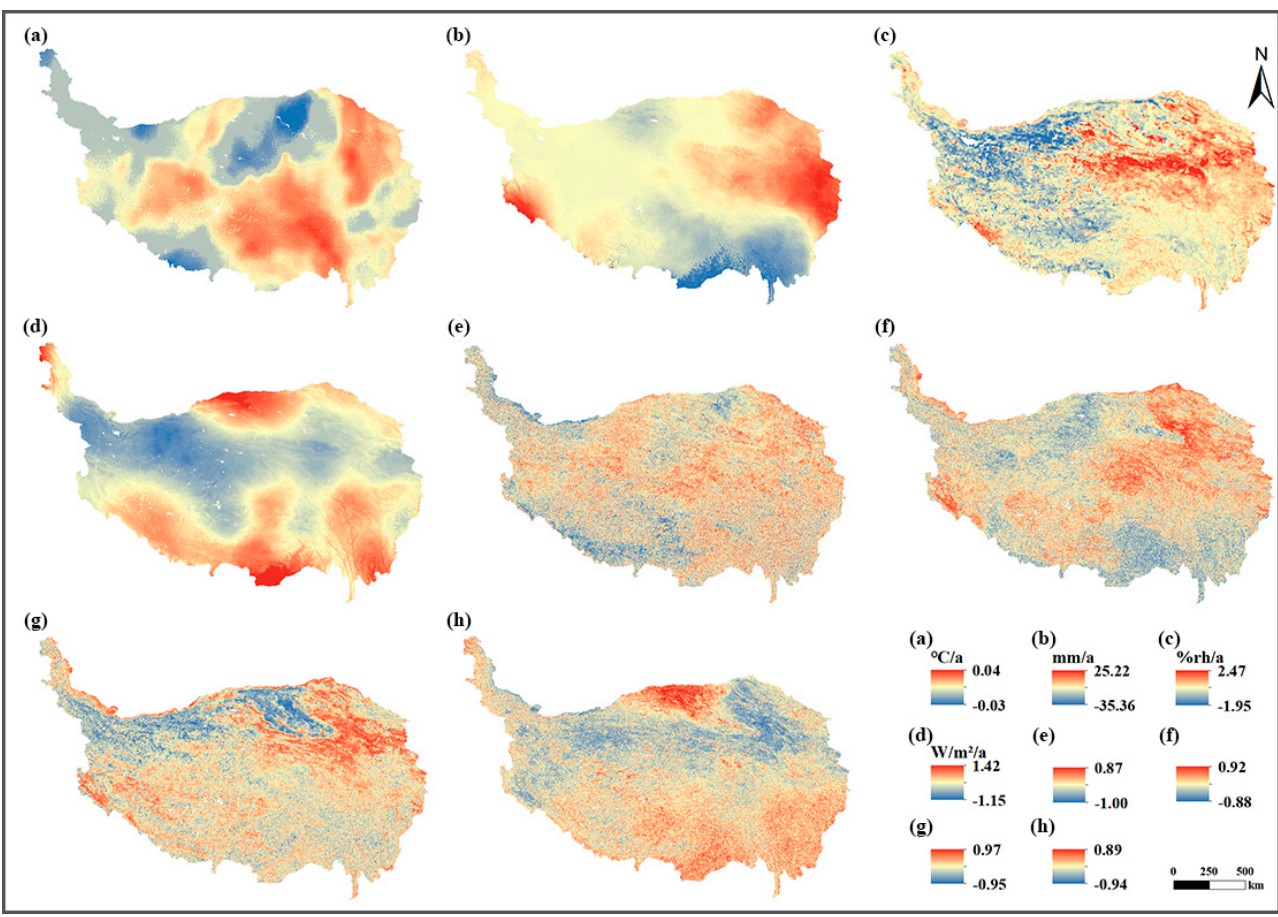

**Figure 5.** Spatial trends (**a**–**d**) of climate factors and their correlation with NDVI (**e**–**h**), including (**a**) temperature, (**b**) precipitation, (**c**) relative humidity, (**d**) solar radiation, (**e**–**h**) correlation coefficient between NDVI and (**e**) temperature, (**f**) precipitation, (**g**) relative humidity, and (**h**) solar radiation.

Regarding humidity, a majority of 60.98% of the plateau exhibited an upward trajectory. The expansion of humidity increase was seen to extend from east to west gradually. The trend of solar radiation on the plateau is generally negatively correlated with the altitude, i.e., solar radiation's trend is more potent in areas with lower altitudes. Over the past 22 years, the solar radiation on the TP has shown a decreasing trend. Merely 16.34% of the region has encountered an augmentation in solar radiation, mainly concentrated in the southern portion of the plateau. The findings suggest that the TP is generally undergoing a "warming and humidification" trend.

To further quantify the influence of climate change on NDVI, a correlation analysis was performed between NDVI and several climate factors from 2000 to 2021. The resulting average correlation coefficients were determined to be 0.314, 0.355, 0.276, and −0.193, respectively (Figure 5e–h). It shows that the correlation between NDVI and precipitation is the largest, with the majority of the TP (accounting for 70.48%) exhibiting a positive correlation. Using the *p*-test method to assess the statistical significance of the correlation results, it was determined that 40.43% of the areas reached a 0.05 significance level. In comparison to the correlation observed between humidity and NDVI, it was determined that the spatial distribution of the two positively correlated response zones exhibited a significant level of consistency. Furthermore, when considering thermal conditions, it is evident that NDVI and temperature show a robust positive correlation. This correlation is particularly pronounced in the southeast TP, which falls within the semi-humid and humid zones. This area is characterized by ample water resources, rendering it highly sensitive to temperature. The relationship between solar radiation and NDVI has a negative correlation. The region exhibiting a negative correlation is situated mainly in the center of the plateau, including approximately 64.38% of the total area. Conversely, the positive correlation area is predominantly found at the edge of the TP. Generally, vegetation growth on the plateau is influenced by two integrated factors: thermal factors (temperature and solar radiation) and moisture factors (precipitation and humidity).

## 5. Discussion

### 5.1. Driving Mechanisms of Vegetation Change

The study found that, from 2000 to 2021, the NDVI on the TP showed little interannual variability and an overall slowly fluctuating upward trend, with an increase rate of 0.015/10a. This was faster than the NDVI change rate in China and is consistent with previous studies on the vegetation coverage variation characteristics in this study area [14]. The results also show that the rate of greening on the TP has increased over the past ten years, which accords with Zhang's forecast based on MODIS NDVI data from 1982 to 2010 [13]. The alpine environment of the TP exhibits a heightened sensitivity to climate changes [55]. The progression towards warming and humidification may have a positive impact on vegetation growth. This will be achieved by mitigating the inhibitory effects of low temperatures on vegetation growth at high altitudes and by augmenting the water supply to vegetation in arid regions [56,57]. Furthermore, some researches have indicated that ecological restoration measures, such as the practice of "Grain for Green", have played a significant role in the revival of vegetation on the TP [58].

In addition, the spatial pattern of NDVI on the TP is not entirely consistent with the distribution of NDVI trends. For instance, the southeast of the plateau, particularly the Hengduan Mountains, is regarded as having a high NDVI (Figure 3). Nevertheless, the growth rate within this region exhibits a notable sluggishness, which can be explained by two factors: First, the vegetation in this area is stable (lower coefficient of variation) and flourishing (greater NDVI), contributing to a modest trend. Second, the Hengduan Mountains have a complex topography and are less affected by human activities [59]. It is evident from the observation above that topography exerts a substantial influence on vegetation growth. The variation of vegetation was primarily influenced by altitude through the regulation of hydrothermal conditions [60]. The slope is vital for surface vegetation material exchange and energy flow, significantly impacting vegetation growth and distribution. A study claims that China's program "Grain for Green" is primarily implemented on slopes higher than 8°, which may help to explain why MO and ST are the predominant terrains showing slight improvement [61].

Meanwhile, consistent with Rumpf's research [60], our study also showed that the rate of warming on the TP increased with elevation, and was faster than that at lower elevations. Vegetation at high altitudes is steadily improving, presumably due to the melting of alpine ice and snow caused by a warming climate, which has shifted the tundra line upwards, expanding the area of ideal growing conditions for vegetation. Furthermore, meltwater

contributes the requisite moisture conditions for vegetation flourishing [62]. It is foreseen that, with global warming, the space for vegetation to survive in the extreme cold zone will be further expanded.

Over the past 22 years, the TP has experienced more significant climate change; hence, it was described as a "driver and amplifier of global change" [6]. Some studies have confirmed that warming and humidification are becoming increasingly significant in northwestern China, and the climate of the arid and semi-arid zone is changing from cold-dry to warm-humid [63]. Random forest models have been widely used in identifying dominant drivers of vegetation change due to their exceptional explanatory capability, rapid computing speed, and robust generalization ability [64]. The climate dominant factor was identified based on the level of relevance after random forests were utilized (Figure 6a). Most of the plateau's NDVI variations are caused by temperature and precipitation change, with the regions dominated by both, making up 80% of the total area (43.39% and 37.23%, respectively). The precipitation-dominated region is mainly located in the central and northeastern parts of the plateau, which is more sensitive to precipitation due to higher temperatures and water evapotranspiration [65]. This region, interestingly, is situated at the border between semi-arid and arid zones, and it also happens to coincide with the medium vegetation coverage strips (Figure 3a). Jenerette's study on the factors influencing vegetation growth in central Asia also revealed that precipitation significantly impacts vegetation at this border [66]. The largest region dominated by temperature is found in the southeast of the TP, with smaller concentrations towards the west. The southeastern region is high in altitude and steep in slope, primarily concentrated in areas where rivers originate. Because of its historically abundant water resources, the temperature has become the dominant factor limiting vegetation growth, and, as a result, it is more sensitive to temperature [67].

Additionally, solar radiation and relative humidity also influence the growth of vegetation. The humidity-dominated region is concentrated in the northeastern TP. The surface soil texture is predominantly sandy and gravelly with poor water retention capacity. Thus, water evaporates more quickly, making vegetation more susceptible to changes in moisture [68]. The region dominated by solar radiation is mainly in the northern TP, an elevated territory known for its frigid climate and perpetual snow and ice coverage. The tundra line shifts upward due to increased solar radiation, enhancing the temperature and light conditions for vegetation growth, especially low-growing plants [65]. Our study also indicated that overall thermal conditions substantially influenced vegetation growth on the TP, especially under extremely topographic conditions (e.g., slope > 5°and altitude > 5000). However, in areas where the topography of the plateau is more moderate (low elevations, gentler slopes), moisture conditions are the dominant climate factor driving vegetation NDVI.

Based on the aforementioned findings, it is evident that temperature and precipitation are the primary climate factors that exert influence on vegetation growth. Furthermore, the vegetation coverage of the TP exhibits a positive trend in response to global warming. On the other hand, ecological restoration measures have been implemented on the TP over the years. These measures have demonstrated a beneficial impact on the improvement of the ecological systems in the region. In this paper, we quantitatively separated the effects of climate change and human activities on NDVI through the residual trend method, and determined the intensity and direction of human activities on vegetation cover based on the positivity and magnitude of the residual trend (Figure 6b–d). Over the past 22 years, there has been a fluctuation in residual values, initially experiencing a decline followed by an upward trajectory. This overall pattern indicates a consistent and progressive increase, suggesting that human activities have had a positive and accelerating effect on vegetation development. In terms of interannual changes, the residual values were small and showed an insignificant decrease from 2000 to 2010, indicating that the impact of human activities on vegetation growth during this period was relatively minor and the negative impacts slightly prevailed. Since 2011, the residual values have increased significantly, and the

positive impacts of human activities on vegetation growth on the plateau have accelerated, which may be attributed to the government's implementation of the policy of returning farmland to grassland and ecological compensation [68].

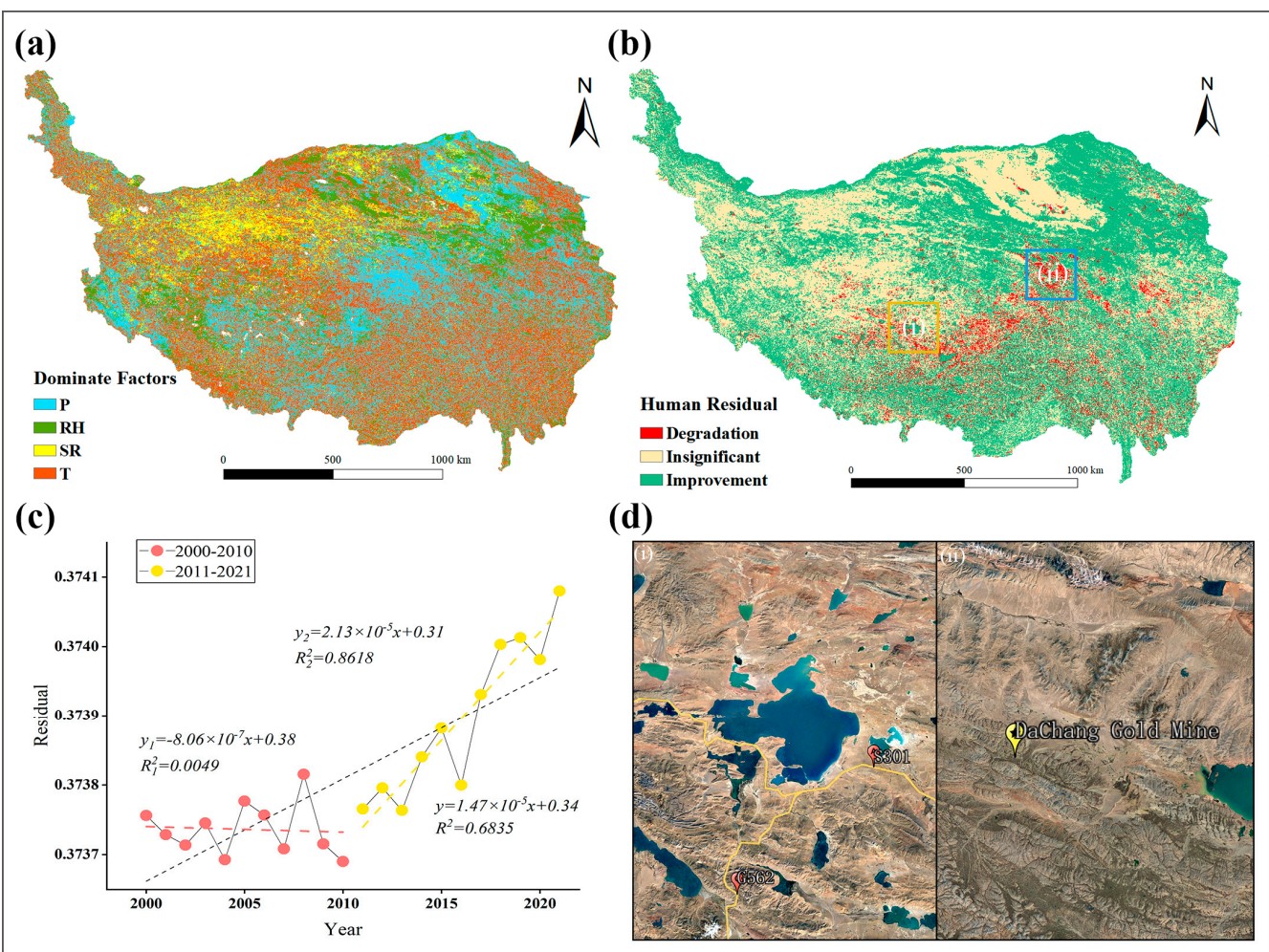

**Figure 6.** Distribution of climate factors that dominate NDVI of vegetation on the Tibetan Plateau (**a**), Spatial and temporal distribution of residual (**b**,**c**). (i) and (ii) in (**b**) correspond to the positions in (**d**) respectively.

Moreover, the residual trend of TP ranged from −0.036 to 0.047/a, and it showed a distinct pattern of "overall growth and local decline", with only a limited number of regions experiencing a significant decline rate (7.98%). Notably, these areas were primarily located in the southwestern Tibetan Plateau. Degraded area (i) is far from human settlements and has many lakes, which is expected to have limited human impacts. Nevertheless, this region is linked to major highways like G562 and G312, and the development of transportation infrastructure has resulted in the deterioration of the local vegetation. Furthermore, site (ii) is situated in Qumalai County, Qinghai Province, home to one of the largest gold mines in Asia. This mine has been found to have a potential resource of 300 tonnes. Mining activities have resulted in the deterioration of the local vegetation. In all, the impact of human activities on vegetation growth is mainly positive.

### 5.2. Superiority of the PCL Model for Simulating NDVI

The NDVI series often exhibit distinct characteristics due to the diversity of spatiotemporal variables and driving factors. The deep learning network's input data are derived from the primary climatic factors that influence the NDVI on the TP, including temperature,

precipitation, humidity, and radiation—and are standardized to enhance the effectiveness of the training process. Our research focuses on multivariate collaborative modeling due to the need to include the influence of meteorological data in simulating NDVI. The datasets were fed into various models to assess accuracy (Figure 7), with RMSE and $R^2$ chosen as the criteria for accuracy evaluation.

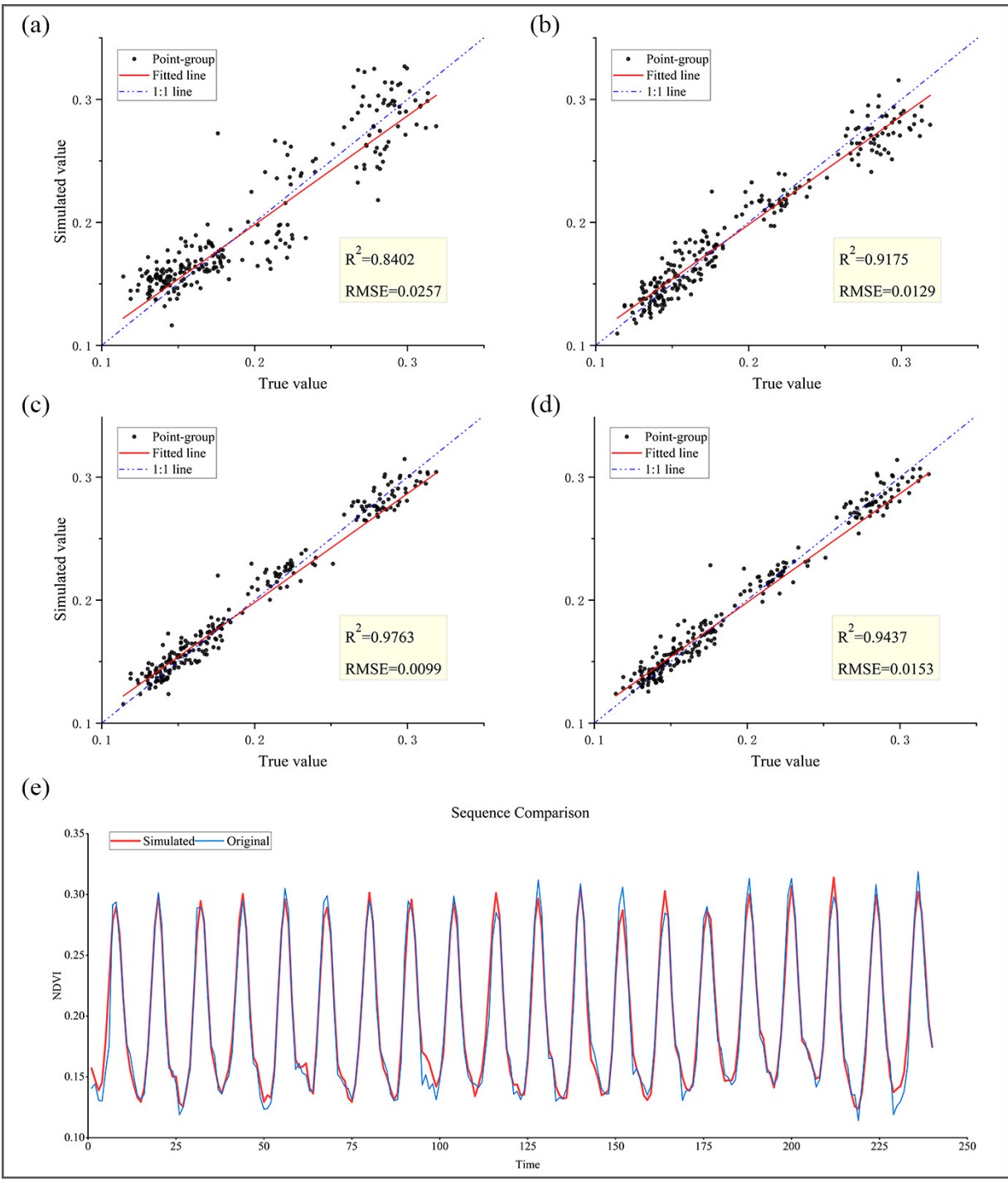

**Figure 7.** The accuracy results of each model for simulating NDVI, from (**a**–**d**) are the simulation results of CNN, LSTM, CNN-LSTM, and PCA-CNN-LSTM(PCL) models, respectively. (**e**) The sequence comparison between the simulated value of PCL and original NDVI series.

The simulation results of the four models and their comparisons with the true values of NDVI are presented in Figure 7. One-dimensional CNN can effectively extract features from time series for prediction purposes. However, it lacks sensitivity to variations in the time series, resulting in a more significant prediction error and outliers (as shown in Figure 7a). Nevertheless, CNN has exceptional performance in extracting hidden spatiotemporal characteristics. The LSTM showed better performance in simulating NDVI (Figure 7b), with an $R^2$ of up to 0.918 and an RMSE of only 0.0129. However, the LSTM network itself is unable to capture spatiotemporal features of the series, and it has to be manually encoded as the input data, which limits the prediction accuracy [69]. In addition, the presence of spatiotemporal features in the sequence during the training of the model leads to a large variance in the adaptive learning rate of the model, which further affects the accuracy of the prediction model. The CNN-LSTM model is devised to forecast time series by leveraging the CNN's ability to extract data feature information and the LSTM's proficiency in processing time series. As shown in Figure 7c, it is evident that the simulated data of CNN-LSTM are highly concentrated and exhibit the maximum level of data correctness, as indicated by an $R^2$ value of 0.976.

Nevertheless, the CNN-LSTM network has a rather intricate topology and suffers from slow computing efficiency due to the substantial volume of input data. Consequently, we enhanced the data input module by utilizing principal component analysis to extract the main components of the original input variables from the feature vectors obtained by the CNN. According to the selection rule of principal components, F1, F2, and F3 were selected as the input principal components. This reduced the data from seven dimensions to three, effectively compressing the input data volume while retaining crucial information. Based on the findings presented in Figure 7d, it is evident that the PCL model, after undergoing parameter optimization, exhibits a high level of prediction accuracy. The evaluation indices demonstrate significant improvement compared to a single model and the predicted outcomes align well with the actual data. It is worth pointing out that although the simulation accuracy of PCL is reduced compared to that of the CNN-LSTM model ($R^2$ decreases by about 3.34%), the computational efficiency is greatly improved due to the reduction of the amount of data inputs. The risk of overfitting is also reduced.

Moreover, it was evident that LSTM has a clear advantage over CNN in simulating NDVI series. Meanwhile, the convolutional kernel pooling function unique to CNN can extract the feature information of the data well. When paired with the memory capabilities of LSTM, it further enhances the simulation performance of the NDVI sequence data. Finally, the model's input and operation efficiency are further optimized for the consideration of principal components. In conclusion, the PCL model has significant superiority in simulating the NDVI series on the Tibetan Plateau.

## 6. Conclusions

Based on the reconstructed NDVI dataset on the GEE, this paper studies the distribution pattern, change characteristics, and influencing mechanisms of NDVI on the Tibetan Plateau. Moreover, we developed a PCA-CNN-LSTM (PCL) model that effectively simulates NDVI.

Temporally, NDVI showed a fluctuating upward trend during the study period, with a growth rate of 0.0134/10a, and the speed accelerated in the last 10 years. Spatially, the vegetation change trend showed a pattern of "general improvement, local stabilization, and degradation located in the southwest". The trend of improvement is most significant in the low altitudes, and the middle altitude is the dominant terrain of degradation. In the future, the improved area is at risk of reversal, which needs to be emphasized.

Over 22 years, there has been a fluctuation in residual values and the positive impacts of human activities on vegetation growth have accelerated since 2011. Among the climate factors, temperature and precipitation play a decisive role in vegetation dynamics, contributing 43.39% and 37.23%, respectively. Above all, vegetation is sensitive to temperature in the western and southeastern regions, and precipitation in the central and northeastern

regions. In addition, our study shows that thermal conditions (temperature and solar radiation) had a more significant influence on vegetation growth on the Tibetan Plateau, especially under extreme topographic zones.

The LSTM model excels at simulating long time series, while the CNN model is adept at extracting hidden spatiotemporal features. Additionally, the data input module is optimized using principal component analysis, which enhances computational efficiency without compromising simulation accuracy. As a result, the PCL model demonstrates a notable advantage in simulating the NDVI on TP.

These findings can give governments more practical evidence to formulate rational ecological conservation policies on the Tibetan Plateau. In the future, spatial and temporal analyses of anthropogenic data (e.g., ecological conservation, human footprint, grazing intensity, etc.) can be collected to give more substantial evidence for quantifying the NDVI trend caused by human activities.

**Author Contributions:** All coauthors made significant contributions to the manuscript. Conceptualization, X.L. and G.D.; data curation, X.L., H.B. and Z.L.; formal analysis, X.L.; funding acquisition, G.D.; investigation, X.L., H.B. and Z.L.; methodology, X.L. and G.D.; project administration, G.D.; resources, G.D.; software, X.L., H.B. and Z.L.; supervision, X.Z.; validation, G.D. and H.B.; visualization, H.B. and X.Z.; writing—original draft, X.L.; writing—review and editing, G.D. All authors have read and agreed to the published version of the manuscript.

**Funding:** This work was funded by the National Program on Key Research Projects of China (no. 2017YFC1502706).

**Data Availability Statement:** The data sources for this paper can be found in Table 1. Other data that support the findings of this study are available from the author, upon reasonable request.

**Acknowledgments:** The authors would like to thank Jianbo Tan from Changsha University of Science and Technology for his guidance on the experimental part of the study. We thank Liu Jinghao for her help in data processing. We also thank the other anonymous reviewers for their constructive comments on the manuscript. Moreover, we thank NASA and TPDC (National Tibetan Plateau Data Centre) for their support of the experimental data.

**Conflicts of Interest:** The authors declare no conflict of interest.

## Appendix A

**Table A1.** Explanation of variance of each component.

| Principal Component | Variance | Variance Contribution Rate/% | Cumulative Variance Contribution Rate/% |
|---|---|---|---|
| F1 | 4.496 | 64.233 | 64.233 |
| F2 | 1.388 | 19.823 | 84.056 |
| F3 | 0.858 | 12.261 | 96.317 |
| F4 | 0.136 | 1.939 | 98.256 |
| F5 | 0,067 | 0.961 | 99.218 |
| F6 | 0.052 | 0.744 | 99.961 |
| F7 | 0.003 | 0.039 | 100.000 |

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
