# Peer review of "Normal Difference Vegetation Index Simulation and Driving Analysis of the Tibetan Plateau Based on Deep Learning Algorithms"

_forests, doi:10.3390/f15010137_

Round 1
Reviewer 1 Report
Comments and Suggestions for Authors
General comments:
The authors have done a great job analysing 22 years of data for a hard-to-reach region. Given the uniqueness of the region, such studies are of great ecological importance. However, the main comment on the manuscript is that the authors do not compare their results with ground-based studies. If ground-based land cover data for the region are unavailable or lacking, it would be desirable to include data from other available satellites, such as Landsat, in the study.
Specific comments:
Line 54. “Monitoring the vegetation index by remote sensing…” Apparently, the manuscript refers to vegetation monitoring using vegetation indices.
Lines 57-58. What are the notable advantages of using NDVI? Compared to what?
Line 60. In the following, the reference to the literary source should be given immediately after the surname of the author.
Lines 66-69. References needed.
Lines 82-83. Reference needed.
Line 90. “…climatic and anthropogenic forces.” There are other important factors such as edaphic and orographic factors.
Lines 109-110. How does this sentence fit with the sentence on Lines 82-83?
Lines 128-130. How was representativeness assessed?
Line 141. There don't seem to be any weather stations in Region D?
Line 150. The abstract says 250m.
Line 172. Table 2. Sources – Specify the date of access to the URL.
Lines 216-218. Commonly known formulae are given.
Lines 283-285. Couldn't this conclusion be reached directly by analysing climatic factors and topography without the mediation of NDVI?
Lines 294-298. Were the VI values corrected depending on the shooting angle?
Lines 304-305. How is the WI value distributed? What is the mean value of the VI - median, arithmetic mean...? Are there statistically significant differences between the WI values by year? For example, between 2000 and 2020.
Lines 306-307. This is a very small range of variation in the VI.
Lines 316-317. “The observed rise in vegetation coverage over the latter decade…” Reference needed.
Lines 324-326. At what level of significance is the null hypothesis rejected?
Lines 395-396. Are these correlation coefficients too low?
Lines 520-523. This conclusion does not follow directly from the results of the study. In our opinion, this paragraph is somewhat declarative in nature. It is desirable to draw such conclusions on the basis of analysing the activities of state environmental authorities, including the state of environmental audits, environmental certification of production, environmental management and marketing, environmental monitoring, environmental education, etc.
Reviewer 2 Report
Comments and Suggestions for Authors
The abstract is comprehensive and includes the results, conclusions and proposed public policies.
The Introduction is quite lengthy, but it lacks references to support some of the statements made. For example, the specific objectives only briefly mention the deep learning techniques that will be employed, without providing sufficient information to inform specialized readers about their advantages and the relevant research supporting their use.
The materials have been described thoroughly, but there are a few details that require attention. For instance, it would be beneficial to include a projection system and Datum in the figures that contain maps. Additionally, it would be useful to evaluate the feasibility of incorporating the information from Table 1 into the legend of Figure 1.
The research methods have been described in detail but they lack a comprehensive explanation of some relevant information that was only cited. This makes it difficult for the reader to replicate the analysis method. One weakness of this section is the processing sequence outlined in Figure 2 which does not match the description in the text. This must be corrected as per the paragraphs just above Figure 2.
All findings (results) were presented satisfactorily, with a clear sequence. The analysis had excellent quality explanatory figures, and Figure 5 was the highlight of the analysis. However, there is a need to review some details regarding certain statements that were not included in the exposition of the results.
During the discussion, it was observed that there are some areas of improvement. One such area is the lack of discussion on the results found based on relevant literature related to the topic. In item 5.1, the paragraph between lines 448-458, both the presentation of results and the methodology used. In item 5.2, there is no discussion about the results presented in Figure 6. To ensure that this article is of high quality and can be published in an international journal, it is important to compare the results, insert relevant concepts that justify the analysis, and turn it into a proper discussion.
The conclusion explains the most relevant findings of the analysis and suggests possibilities for future research based on the obtained results.
All of these details are in the attached document as comments and should be considered.

Reviewer 3 Report
Comments and Suggestions for Authors
The paper is interesting and deals with an important topic. I would suggest just to write some sentences on why the Authors have chosen the Random Forest Algorithm and what they are expecting from it.
Reviewer 4 Report
Comments and Suggestions for Authors
line 207: Whether the Pirson correlation coefficient is suitable for the short data series of n=22? Why did you not employed nonparametric correlation?
lines 214-218, or formulas (5)-(8): Why is the same data series (1,2,3,...,n) marked with the two different symbols (t vs. τ)? In this concept, what is τ? This question has appeared because unclear is the following: 1) in formula (7), whether only one value of U(t,τ) is calculated or several different values (for each τ) are? 2) in formula (8), how can there be a maximum and a minimum for one calculated value of U(t,τ)?
lines 220-225 vs. line 333 ("The average Hurst exponent of the TP was 0.43"): Is there any significance test for Hurst exponent values to check their difference from 0.5?
lines 452-454: "Based on a thorough analysis of data volume and computation rate, we use 10-fold cross-validation, where the training set is divided into 10 equal parts, 9 of which are used as the training set and the remaining as the validation set." This should be just explained in Section 3.5.
lines 533-534: "This paper uses 70% of the NDVI data as the training dataset and allocates the remaining 30% as the test dataset incorporating the aforementioned studies." This also should be just explained in Section 3.5.
Some technical notes:
lines 128-129: To reach better understanding of the whole paragraph sense, the following edition can be proposed - "Five representative regions (Figure 1, A-E) have been chosen based on a detailed analysis of topography, climate ..."
lines 135-136: For the same purpose, also the following edition would be useful - "the Himalayas and the Qaidam basin (region E)."
lines 138-139 and further: Also useful would be to specify everywhere like these - "the Hengduan Mountain (region B), "the Qaidam Basin (region E)", and "the Northern Tibetan Plateau (region D)", etc.
Table 1: Here, "Area percentage" (like in Table 3) would be clearer than "Ratio"
Comments on the Quality of English Languageline 23: "TP saw ... trend"? Do you mean "TP showed ... trend" or "TP demonstrated ... trend"?
line 75: Here, the word "changes" would be more exact than "alterations"
line 89: Here, "factors" would be more suitable word than "elements"
lines 195-196: "Additionally, employing the coefficient of variation (CV) to quantify the stability of the NDVI series". This sentence is constructed without a subject and predicate.
line 285: In this phrase, enough is only one of the two words: "consequently" or "subsequently".
lines 288-291: Here, the same word "slope" is used in two different senses: 1) as "aspect" (see Fig. 3d) that means geographical azimuth and here applies to the cases SHA, HSH, SUN, HSU; 2) as "incline" that means steepness of a surface and is more common for the word "slope", and here applies to the case FLS. To make the discourse clearer, recommended is to write "aspect" (instead "slope") when implying light and warm availability on surface. Then the last case can be explained as "flat surface without specific aspect (FLS)".
